# Meander-Line Slot-Loaded High-Sensitivity Microstrip Patch Sensor Antenna for Relative Permittivity Measurement

**DOI:** 10.3390/s19214660

**Published:** 2019-10-27

**Authors:** Junho Yeo, Jong-Ig Lee

**Affiliations:** 1School of Computer and Communication Engineering, Daegu University, 201 Daegudae-ro, Gyeongsan, Gyeongbuk 38453, Korea; 2Department of Electronics Engineering, Dongseo University, San69-1, Jurye-2dong, Sasang-gu, Busan 47011, Korea; leeji@dongseo.ac.kr

**Keywords:** meander-line slot, high-sensitivity, microstrip patch sensor antenna, permittivity measurement

## Abstract

A high-sensitivity microstrip patch sensor antenna (MPSA) loaded with a meander-line slot (MLS) is proposed for the measurement of relative permittivity. The proposed MPSA was designed by etching the MLS along the radiating edge of the patch antenna, and it enhanced the relative permittivity sensitivity with an additional effect of miniaturization in the patch size by increasing the slot length. The sensitivity of the proposed MPSA was compared with that of a conventional rectangular patch antenna and a rectangular slit (RS)-loaded MPSA, by measuring the shift in the resonant frequency of the input reflection coefficient. Three MPSAs were designed and fabricated on a 0.76 mm-thick RF-35 substrate to resonate at 2.5 GHz under unloaded conditions. Sensitivity comparison was performed by using five different standard dielectric samples with dielectric constants ranging from 2.17 to 10.2. The experiment results showed that the sensitivity of the proposed MPSA is 6.84 times higher for a low relative permittivity of 2.17, and 4.57 times higher for a high relative permittivity of 10.2, when compared with the conventional MPSA. In addition, the extracted relative permittivity values of the five materials under tests showed good agreement with the reference data.

## 1. Introduction

Accurate characterization of the complex permittivity of a material is very important in various applications, such as wireless communication, biomedical, healthcare, agriculture, chemistry, and the food industry [1]. There exist various permittivity measurement methods, and they are classified into non-resonant and resonant methods. Recently, the resonant methods with planar resonators, such as split ring resonator (SRR) and complementary SRR (CSRR) structures, have been widely used because of their simple geometry, small size, ease of fabrication, and low cost [2,3,4]. Microstrip structures have been extensively applied for complex permittivity measurement because of their advantages compared to other structures such as low weight, low profile, low cost, conformance to platform surface, ease of fabrication, and compatibility with integrated circuits. SRR-based and CSRR-based sensors in the microstrip transmission line as a band stop filter have been extensively investigated [5,6,7,8,9,10,11,12,13,14,15,16,17,18]. An interdigital capacitor-based SRR structure coupled with a microstrip transmission line was used as a microstrip resonant sensor at 2.45 GHz to enhance the relative permittivity sensitivity of dielectric materials [19]. A CSRR structure based on an interdigital-capacitor-shaped defected ground structure was introduced at 1.5 GHz for high-sensitivity relative permittivity characterization [20]. The measured sensitivity of the proposed sensor was two times higher for a low relative permittivity material at 2.17, and it was 1.42 times higher for a high relative permittivity material at 10.2, when compared with the conventional double-ring CSRR-based sensor.

In recent years, the microstrip antenna has been researched as a wireless sensor for temperature, strain, and permittivity measurement because it can be considered a resonator with the capability of communication. The temperature–resonant frequency relationship of a rectangular microstrip patch antenna considering the substrate dielectric constant and the base metallic material bonded on the ground plane was derived [21]. Temperature sensing without electronics was demonstrated through the far-field wireless interrogation of a microstrip patch antenna sensor [22]. The relationship between the normalized frequency shift of the antenna sensor and temperature changes up to 280 °C was derived and it was verified experimentally. A rectangular microstrip patch antenna was used as the strain sensing element of a real-time wireless vibratory strain sensing system with a dynamic interrogator using a frequency-modulated continuous wave radar [23]. The strain directional characteristics for the two resonant frequencies of a circular microstrip patch antenna were analyzed [24]. A dual-band microstrip patch antenna printed on a flexible felt substrate was designed as a strain sensor for structural health-monitoring applications [25]. The resonant frequency of a microstrip patch antenna covered with a dielectric layer accounting for a coated protective layer, an ice layer, or a layer for plasma contact was predicted by using the effective dielectric constant of the whole structure, which can be calculated by using the variational technique [26]. A theoretical technique for accurate analysis of the input impedance and resonant frequency of probe-fed ordinary and stacked microstrip patch antennas covered with a dielectric material was proposed based on the spectral domain Green’s function [27]. An air-spaced coaxial-fed rectangular microstrip patch antenna was used as a sensor for relative permittivity measurement of solid materials such as polytetrafluoroethylene, Thordon, nylon, and acrylic when they are considered as a superstrate [28,29]. For liquid materials such as cyclohexane, pentanol, butan-1-ol, ethanol, and water, the microstrip patch is considered to be buried in the liquid materials, and an effective dielectric constant was derived by using them in the buried microstrip structure. Moisture content ranging from 0% to 30% in peat and loam soil was measured by using single- and dual-resonant microstrip patch antenna sensors [30]. A CSRR-loaded microstrip patch antenna was proposed as a microfluidic ethanol chemical sensor to detect ethanol concentrations that varied from 0% to 100% [31]. The salinity in seawater was determined by measuring the shift in the resonant frequency of a microstrip patch sensor with a liquid chamber in the substrate based on the relative permittivity change of seawater for different levels of salinity [32]. A method to determine the dielectric constant and loss tangent of dielectric materials using the change in the input impedance of a microstrip patch antenna was proposed when the dielectric sample in free space is illuminated by an incident field from an antenna placed nearby [33]. Recently, a rectangular microstrip patch antenna loaded with a thin rectangular slot along the radiating edge of the patch at 2.5 gigahertz (GHz) was introduced for high-sensitivity relative permittivity characterization [34]. The measured sensitivity of the slot-loaded patch antenna was 3.54 to 4.53 times higher than the conventional patch antenna for five dielectric samples with relative permittivity ranging from 2.17 to 10.2.

Recently, a meander-line slot (MLS) has been applied to microstrip patch antennas for miniaturization and multi-band operation. The resonant frequencies, bandwidth, and gain characteristics of a microstrip patch antenna were compared when single and dual MLSs were loaded in various locations in the patch [35]. A miniaturized tri-band microstrip patch antenna loaded with an MLS in the middle of the patch and a double circular ring resonator in the ground was proposed for operating at 3.2 GHz, 5.4 GHz, and 5.8 GHz [36]. However, an attempt to improve the relative permittivity sensitivity by using the MLS for the microstrip patch antenna has not been tried. 

In this paper, an MLS-loaded high-sensitivity microstrip patch sensor antenna (MPSA) is proposed for relative permittivity characterization of lossless and low loss dielectric materials. The MLS was designed to miniaturize the patch size and enhance the relative permittivity sensitivity of the proposed MPSA. The sensitivity of the proposed MLS-loaded microstrip patch antenna was compared with a conventional patch antenna and an MPSA loaded with a rectangular slit (RS). The sensitivity was measured by the shift in the resonant frequency of the input reflection coefficient (S_11_) when the planar dielectric material under test (MUT) was placed above the patch as a superstrate. The three MPSAs were designed and fabricated on a 0.76-mm-thick RF-35 substrate for the first resonant frequency at 2.5 GHz under unloaded conditions. Full-wave simulations were performed using CST Microwave Studio.

## 2. MLS-Loaded MPSA Design

Figure 1 shows the geometries of a conventional inset-fed rectangular MPSA, an RS-loaded MPSA, and the proposed MLS-loaded MPSA. The corresponding S_11_ characteristics of the three MPSAs are shown in Figure 2.

The conventional inset-fed rectangular MPSA was designed to resonate at *f*_r1_ = 2.5 GHz under unloaded conditions by using the equations in [37] with an RF-35 substrate (*ε*_r_ = 3.5, tan *δ* = 0.0018, *h* = 0.76 mm), as shown in Figure 1a [34]. The calculated width and length of the rectangular patch were *W*_1_ = 40.0 mm and *L*_1_ = 31.9 mm, respectively. The width and length of the 50 ohm microstrip feed line were *w*_f1_ = 1.66 mm and *l*_f1_ = 24.5 mm, respectively, whereas those of the inset were *w*_is1_ = 2.8 mm and *l*_is1_ = 9 mm, respectively. The width and length of the ground plane are *W*_g_ = *L*_g_ = 80 mm. The MUT will be placed above the patch as a superstrate for loaded conditions, as shown in Figure 1a. Two other higher-order resonant frequencies exist at *f*_r2_ = 4.03 GHz and *f*_r3_ = 4.584 GHz.

Next, an RS was etched along the radiating edge of the patch on the opposite side of the microstrip feed line, as shown in Figure 1b, and two resonant frequencies were generated at *f*_r1_ = 2.5 GHz and *f*_r2_ = 3.465 GHz with a frequency ratio of *f*_r2_/*f*_r1_ = 1.39 [34,38]. The frequency bandwidth of the first resonant frequency for a voltage standing wave ratio (VSWR) < 2 was 2.496–2.503 GHz (0.28%), which is quite reduced compared to the conventional MPSA (2.490–2.510 GHz, 0.8%) because of the size reduction of the RS-loaded patch. The third resonant frequency, *f*_r3_, existed slightly above 5 GHz. The width and length of the ground plane and the width of the feed line were the same as the conventional MPSA in Figure 1a. The width and length of the rectangular patch were scaled down to *W*_2_ = 31.8 mm and *L*_2_ = 25.4 mm, respectively, in order to make the first resonant frequency 2.5 GHz. The size reduction was about 20.4%, compared to the conventional MPSA. The length of the feed line was slightly increased to *l*_f2_ = 27.3 mm. The width and length of the RS were *w*_rs1_ = 1 mm and *l*_rs1_ = 29.8 mm, respectively. The distance between the radiating edge and the thin rectangular slot was *w*_o1_ = 1 mm. The width and length of the inset were *w*_is2_ = 2.3 mm and *l*_is2_ = 12 mm, respectively. Note that the location of the RS was chosen because the electric field concentration on the patch was highest at this position as shown in Figure 3b. 

Thirdly, an MLS was loaded along the radiating edge of the patch, as shown in Figure 1c. The shape of the proposed MLS looked like a single-rectangular-ring CSRR with L-shaped slots appended at both sides of the split gap, and its total length was much longer than that of the RS. In this case, the frequency bandwidth of *f*_r1_ for a VSWR < 2 was further reduced to 2.498–2.502 GHz (0.16%) with an increased quality factor. The dimensions of the patch were scaled down to *W*_3_ = 22.0 mm and *L*_3_ = 17.6 mm, which were reduced by about 44.9% compared to the conventional MPSA. A quarter-wavelength transformer was used on the feed line to match the 50 ohm input impedance. The dimensions of the MLS and other geometric parameters to make the first resonant frequency 2.5 GHz were as follows: *w*_rs2_ = 1 mm, *l*_rs2_ = 20.0 mm, *w*_o2_ = 1 mm, *g*_1_ = 1 mm, *l*_rs3_ = 9.5 mm, *g*_2_ = 1 mm, *l*_rs4_ = 9.5 mm, *w*_qt_ = 0.6 mm, *l*_qt_ = 18.3 mm, *w*_is3_ = 1.5 mm, *l*_is3_ = 4 mm, *w*_f3_ = 1.66 mm, *l*_f3_ = 31.2 mm, and *W*_g_ = *L*_g_ = 80 mm. In this case, the second resonant frequency was *f*_r2_ = 3.233 GHz, located relatively close due to the increased slot length compared to the RS-loaded case. The third resonant frequency existed at *f*_r3_ = 4.584 GHz. The relationship between the first resonant frequency and the geometric parameters of the MLS could be derived based on the resonant frequency variation obtained from the simulation, and it was as follows:(1)$$f_{r1}=\frac{c}{2\sqrt{\varepsilon_{reff}}\left(L_{3}+2\Delta L+a_{0}+a_{1}l_{rs2}+2a_{2}g_{1}+2a_{3}l_{rs3}+2a_{4}g_{2}+2a_{5}l_{rs4}\right)}$$
where *L*_3_ is the length of the rectangular patch, *f*_r1_ is the first resonant frequency, *c* is the speed of light in free space, *ε*_reff_ is the effective relative permittivity, Δ*L* is the extension of the patch length due to the fringing effects, *a*_0_ is the coefficient to compensate for the shift in the first resonant frequency due to the inset between the patch and the quarter-wavelength transformer, *a*_1_ is the coefficient related to *l*_rs2_, *a*_2_ is the coefficient related to 2*g*_1_, *a*_3_ is the coefficient related to 2*l*_rs3_, *a*_4_ is the coefficient related to 2*g*_2_, and *a*_5_ is the coefficient related to 2*l*_rs4_. *ε*_reff_ and Δ*L* can be calculated by using the equations in [37], which are *ε*_reff_ = 3.3 and Δ*L* = 0.3538 mm. The calculated values of the coefficients were as follows: *a*_0_ = 0.4248ⅹ10^−3^, *a*_1_ = 0.2355, *a*_2_ = 0.5627, *a*_3_ = 0.1511, *a*_4_ = 0.3031, and *a*_5_ = 0.2612.

Figure 3 shows the electric field distributions of the conventional, the RS-loaded, and the proposed MLS-loaded MPSAs at the first resonant frequency of 2.5 GHz. We can see that the electric fields are widespread along the two radiating edges of the patch for the conventional MPSA. When RS is loaded along one of the radiating edges, the electric fields are concentrated along the RS, as shown in Figure 3b. For the proposed MLS-loaded MPSA, the electric fields were more concentrated near the MLS with the reduced area. Therefore, the size of the MUT can be extensively reduced, compared to the conventional MPSA.

## 3. Sensitivity Comparison

In this section, the sensitivity of the proposed MLS-loaded MPSA is compared with the conventional and the RS-loaded MPSAs by measuring the shift in the first resonant frequency of the input reflection coefficient. 

Figure 4 shows the S_11_ characteristics of the conventional, the RS-loaded, and the proposed MLS-loaded MPSAs when the MUT was placed above the patch as a superstrate. The MUT’s relative permittivity (*ε*_r_) was varied from 1 to 10 in increments of 1 with a zero loss-tangent, as in the lossless case. The width and length of the MUT are the same as those of the ground plane, and its thickness was chosen to be 1.6 mm because the thickest of the substrates available from Taconic Inc., which were used as the MUTs, was around 1.6 mm. For the conventional MPSA, the first resonant frequency of S_11_ moves from 2.5 GHz to 2.312 GHz when *ε*_r_ increases from 1 to 10, whereas it shifts from 2.5 GHz to 1.854 GHz for the RS-loaded MPSA. The first resonant frequency of the proposed MLS-loaded MPSA moves from 2.5 GHz to 1.585 GHz when *ε*_r_ increases from 1 to 10. Table 1 summarizes the first resonant frequencies of the S_11_ responses with a different relative permittivity of the MUT for the three MPSAs.

In order to compare the sensitivity of the proposed MPSA with that of the conventional and the RS-loaded MPSAs, we used the following definitions [2,9,10,20,31,34]:(2)$$\Delta f_{r}=f_{ru}-f_{rl}\;(GHz)$$
(3)$$PRFS=\frac{\Delta f_{r}}{f_{r}}=\frac{f_{ru}-f_{rl}}{f_{ru}}\times 100(\%)$$
(4)$$PRFSE=\frac{PRFS_{slot\text{-}loaded\;MPSA}}{PRFS_{conventional\;MPSA}}$$
(5)$$S=\frac{\Delta f_{r}}{\Delta \varepsilon_{r}}=\frac{f_{ru}-f_{rl}}{\varepsilon_{ru}-\varepsilon_{rl}}$$
(6)$$SE=\frac{S_{slot\text{-}loaded\;MPSA}}{S_{conventional\;MPSA}}$$
where Δ*f*_r_ is the shift in the first resonant frequency of the three MPSAs, *f*_ru_ is the first resonant frequency of the three MPSAs for unloaded conditions, *f*_rl_ is the first resonant frequency of the three MPSAs for loaded conditions, PRFS is the percent relative frequency shift of the first resonant frequency of the three MPSAs, PRFSE is the enhancement in the percent relative frequency shift of the first resonant frequency of the RS-loaded and the proposed MLS-loaded MPSAs, compared to the conventional MPSA, S is the sensitivity of the first resonant frequency of the three MPSAs with respect to the relative permittivity of the MUT, *ε*_ru_ is the relative permittivity of the MUT for unloaded conditions, *ε*_rl_ is the relative permittivity of the MUT for loaded conditions, and SE is the sensitivity enhancement of the first resonant frequency of the RS-loaded and the proposed MLS-loaded MPSAs compared to the conventional MPSA.

Based on the results shown in Figure 4 and Table 1, the sensitivity performance of the three MPSAs was analyzed using Equations (2)–(6). First, Δ*f*_r_ of S_11_ for the three MPSAs is plotted in Figure 5a. Figure 5b shows the PRFS of the three MPSAs. The PRFSE for the RS-loaded and the proposed MLS-loaded MPSAs, compared to the conventional MPSA, is presented in Figure 5c. Next, S of the first resonant frequencies for the three MPSAs is plotted in Figure 5d. Figure 5e presents the SE of the RS-loaded and the proposed MLS-loaded MPSAs, compared to the conventional MPSA.

We observe from Figure 5d that sensitivity S is not constant as a function of relative permittivity. For example, when the permittivity of the MUT is ε_r_ = 2, frequency shift Δ*f*_r_ in the first resonant frequency of the conventional MPSA is 0.032 GHz, whereas for the RS-loaded and the proposed MLS-loaded MPSAs it is 0.116 GHz and 0.187 GHz, respectively. PRFS in the first resonant frequency of the conventional MPSA is 1.28%, whereas for the RS-loaded and the proposed MLS-loaded MPSAs it is 4.64% and 7.48%, respectively. Therefore, PRFSE for the RS-loaded and the proposed MLS-loaded MPSAs, compared to the conventional MPSA, are 3.63 and 5.84. In this case, S is the same as Δ*f*_r_ because the difference in the relative permittivity (Δε_r_) is 1. SE values of the first resonant frequency of the RS-loaded and the proposed MLS-loaded MPSAs, compared to the conventional MPSA, are 3.63 and 5.84. It is worthwhile to note that PRFSE is the same as SE, and it can be used to estimate the sensitivity enhancement when the resonant frequencies of the MPSAs are different, because the sensitivity enhancement cannot be used for different resonant frequencies.

When the permittivity of the MUT is increased to *ε*_r_ = 3, Δ*f*_r_ in the first resonant frequency of the conventional MPSA is 0.056 GHz, whereas those for the RS-loaded and the proposed MLS-loaded MPSAs are 0.213 GHz and 0.336 GHz, respectively. PRFS in the first resonant frequency of the conventional MPSA is 2.24%, whereas for the RS- and the proposed MLS-loaded MPSAs, they are 8.52% and 13.44%, respectively. Therefore, PRFSE values for the RS-loaded and the proposed MLS-loaded MPSAs, compared to the conventional MPSA, are 3.80 and 6.00. S values for the first resonant frequency of the conventional, the RS-loaded, and the proposed MLS-loaded MPSAs are 0.028 GHz, 0.107 GHz, and 0.168 GHz, respectively. SE values of the first resonant frequency of the RS-loaded and the proposed MLS-loaded MPSAs compared to the conventional MPSA, are 3.80 and 6.00. 

As the permittivity of the MUT is further increased to *ε*_r_ = 10, Δ*f*_r_ in the first resonant frequency of the conventional MPSA is 0.188 GHz, whereas for the RS-loaded and the proposed MLS-loaded MPSAs, they are 0.646 GHz and 0.915 GHz, respectively. PRFS in the first resonant frequency of the conventional MPSA is 7.52%, whereas for the RS-loaded and the proposed MLS-loaded MPSAs, they are 25.84% and 36.60%, respectively. Therefore, PRFSE for the RS-loaded and the proposed MLS-loaded MPSAs, compared to the conventional MPSA, are 3.44 and 4.87. S values for the first resonant frequency of the conventional, the RS-loaded, and the proposed MLS-loaded MPSAs are 0.021 GHz, 0.072 GHz, and 0.102 GHz, respectively. SE values for the first resonant frequency of the RS-loaded and the proposed MLS-loaded MPSAs, compared to the conventional MPSA, are 3.44 and 4.87. Hence, we can conclude that the sensitivity of the proposed MLS-loaded MPSA at the first resonant frequency is 4.87 to 6.00 times higher than the conventional MPSA for relative permittivity values ranging from 2 to 10.

Next, in order to determine the relationship between PRFS of the first resonant frequency in the proposed MLS-loaded MPSA and the relative permittivity of the MUT, a curve-fitting tool in SigmaPlot was used and a fifth-order polynomial function was chosen for the fitting function.
(7)$$\begin{array}{l} \varepsilon_{r}=1+0.12\times \mathrm{PRFS}+1.3475\times 10^{-3}\times {\mathrm{PRFS}}^{2}+7.5954\times 10^{-5}\times {\mathrm{PRFS}}^{3}\\ \;-1.727\times 10^{-6}\times {\mathrm{PRFS}}^{4}+3.3143\times 10^{-8}\times {\mathrm{PRFS}}^{5} \end{array}$$

The PRFS of the simulated first resonant frequency for the proposed MPSA when the MUT’s relative permittivity (*ε*_r_) was varied from 1 to 10 in increments of 1 was used to derive Equation (7). Figure 6 compares the simulated and curve-fitted relative permittivity of the MUT as a function of PRFS. Errors between the simulated and curve-fitted relative permittivity at the input data points were kept to less than 0.1%.

## 4. Experiment Results and Discussion

Prototypes of the three MPSAs were fabricated on an RF-35 substrate (*ε*_r_ = 3.5, tan *δ* = 0.0018, *h* = 0.76 mm), as shown in Figure 7. The S_11_ characteristics of the fabricated antenna were measured using an Agilent N5230A network analyzer, and photographs of the experiment setup are shown in Figure 8. Five different standard dielectric MUTs from Taconic, Inc., were tested for the sensitivity comparison. The MUTs had relative permittivity ranging from 2.17 to 10.2, and their relative permittivity, loss tangent, and thickness from the data sheet in [39] are summarized in Table 2. 

First, the S_11_ characteristics of the three MPSAs were simulated with the five MUTs in Table 2, as shown in Figure 9. The length of the MUTs was slightly reduced to 75 mm because there is a protruding part in the SMA connector for soldering the input port of the fabricated antenna, as shown in Figure 8b. For the conventional MPSA, the first resonant frequency for S_11_ moved from 2.465 GHz for the TLY-5A MUT (*ε*_r_ = 2.17) to 2.312 GHz for the RF-10 MUT (*ε*_r_ = 10.2). In the RS-loaded MPSA, the first resonant frequency shifted from 2.369 GHz for TLY-5A to 1.852 GHz for RF-10, whereas it moved from 2.285 GHz for TLY-5A to 1.577 GHz for RF-10 in the proposed MLS-loaded MPSA. Table 3 shows the first resonant frequencies of the simulated S_11_ responses for the three MPSAs.

Figure 10 compares the simulated sensitivity for the first resonant frequencies of the three MPSAs when the five MUTs were loaded. When TLY-5A was used as the MUT, the first resonant frequency shifts Δ*f*_r_ for the conventional, the RS-loaded, and the proposed MLS-loaded MPSAs were 0.035 GHz, 0.131 GHz, and 0.215 GHz, respectively, and PRFS for each of them were 1.40%, 5.24%, and 8.60%, respectively. PRFSE values for the first resonant frequency of the RS-loaded and the proposed MLS-loaded MPSAs, compared to the conventional MPSA, ware 3.74 and 6.14, respectively. S for the conventional, the RS-loaded, and the proposed MLS-loaded MPSAs were 0.030 GHz, 0.112 GHz, and 0.184 GHz, respectively. SE values of the RS-loaded and the proposed MLS-loaded MPSAs, compared to the conventional MPSA, were 3.74 and 6.14. When RF-10 was used as the MUT, the first resonant frequency shifts Δ*f*_r_ for the conventional, the RS-loaded, and the proposed MLS-loaded MPSAs were 0.188 GHz, 0.648 GHz, and 0.923 GHz, respectively, while PRFS was 7.52%, 25.92%, and 36.92%, respectively. PRFSE values for the first resonant frequency of the RS-loaded and the proposed MLs-loaded MPSAs, compared to the conventional MPSA, were 3.45 and 4.91, respectively. S for the conventional, the RS-loaded, and the proposed MLS-loaded MPSAs were 0.020 GHz, 0.070 GHz, and 0.100 GHz, respectively. SE values for the RS-loaded and the proposed MPSAs, compared to the conventional MPSA, were 3.45 and 4.91. Therefore, the trends in Δ*f*_r_, PRFS, PRFSE, S, and SE for the five MUTs are similar to those of the lossless case in Figure 5.

Next, the S_11_ characteristics of the three MPSAs were measured in order to validate the simulated sensitivity for the five MUTs described in Table 2, as shown in Figure 11. The length of the MUTs was slightly reduced to 75 mm. When unloaded, the first S_11_ resonant frequencies of the conventional, the RS-loaded, and the proposed MLS-loaded MPSAs were 2.528 GHz, 2.509 GHz, and 2.506 GHz, respectively. The respective errors compared to the simulated first resonant frequency when unloaded were 1.12%, 0.36%, and 0.24%, which might have been caused by errors in fabrication and measurement. For the conventional MPSA, the first resonant frequency of S_11_ moved from 2.497 GHz for TLY-5A to 2.328 GHz for RF-10, while it moved from 2.376 GHz for TLY-5A to 1.835 GHz for RF-10 in the RS-loaded MPSA. In the proposed MLS-loaded MPSA, the first resonant frequency of S_11_ moved from 2.294 GHz for TLY-5A to 1.592 GHz for RF-10. We note that the S11 magnitude levels are larger than −10 dB due to impedance mismatch for the proposed MLS-loaded MPSA, but its effect on the first resonant frequency and the sensitivity is very little. Measured first resonant frequencies of the three MPSAs are summarized in Table 4. 

The measured sensitivities for the first resonant frequencies of the three MPSAs are compared in Figure 12. When TLY-5A (*ε*_r_ = 2.17), RF-301 (*ε*_r_ = 2.97), TRF-43 (*ε*_r_ = 4.3), RF-60A (*ε*_r_ = 6.15), and RF-10 (*ε*_r_ = 10.2) were used as the MUT, in turn, the first resonant frequency shifts, Δ*f*_r_, of the conventional MPSA were 0.031 GHz, 0.053 GHz, 0.083 GHz, 0.134 GHz, and 0.200 GHz, respectively, whereas those of the RS-loaded MPSA were 0.133 GHz, 0.208 GHz, 0.322 GHz, 0.449 GHz, and 0.674 GHz, respectively. In the proposed MLS-loaded MPSA, Δ*f*_r_ was 0.212 GHz, 0.322 GHz, 0.482 GHz, 0.678 GHz, and 0.914 GHz, respectively. PRFS values for the conventional MPSA were 1.23%, 2.10%, 3.28%, 5.30%, and 7.91%, respectively, whereas the RS-loaded MPSA showed 5.30%, 8.29%, 12.83%, 17.89%, and 26.86%, respectively. In the proposed MLS-loaded MPSA, PRFS values were 8.46%, 12.85%, 19.23%, 27.06%, and 36.47%, respectively. PRFSE values for the first resonant frequency of the RS-loaded MPSA were 4.32, 3.95, 3.91, 3.38, and 3.40, respectively, whereas they were 6.90, 6.13, 5.86, 5.10, and 4.61, respectively, for the proposed MLS-loaded MPSA. S values in the conventional MPSA were 0.027 GHz, 0.027 GHz, 0.025 GHz, 0.026 GHz, and 0.022 GHz, respectively, whereas those in the RS-loaded MPSA were 0.114 GHz, 0.106 GHz, 0.098 GHz, 0.087 GHz, and 0.073 GHz, respectively. In the proposed MLS-loaded MPSA, S values were 0.181 GHz, 0.164 GHz, 0.146 GHz, 0.132 GHz, and 0.100 GHz, respectively. SE values for the RS-loaded MPSA were 4.29, 3.92, 3.88, 3.35, and 3.37, respectively, whereas they were 6.84, 6.08, 5.81, 5.06, and 4.57, respectively, for the proposed MLS-loaded MPSA. Hence, the measured sensitivity of the proposed MLS-loaded MPSA is 4.57 to 6.84 times higher than the conventional MPSA for the five MUTs, with relative permittivity ranging from 2.17 to 10.2. We note that the measured SE value of the proposed MPSA for low permittivity MUTs such as TLY-5A, RF-301, and RF-60A increases compared to the simulated one, whereas it decreases for the other high permittivity MUTs.

In order to validate the performance of the proposed MLS-loaded MPSA, the relative permittivity of the five MUTs were extracted by using the measured PRFS and Equation (7), and the results are compared in Table 5. It is observed that the absolute value of the maximum error between the extracted and the reference relative permittivity ranges between 0.91% and 2.68%, which is within the tolerance provided by Taconic, Inc. [39]. The measured errors result from errors in fabrication and measurement.

## 5. Conclusions

In this paper, a new, high-sensitivity MPSA has been proposed by using an MLS loaded along the radiation edge of the patch for relative permittivity measurement at 2.5 GHz. The sensitivity of the proposed MLS-loaded MPSA was compared with that of a conventional MPSA without the slot, and compared with an RS-loaded MPSA by measuring the shift in the first resonant frequency of the input reflection coefficient when the planar MUT was placed above the patch. The measured sensitivity of the proposed MPSA is 4.57 to 6.84 times higher than the conventional MPSA for the five MUTs with a relative permittivity ranging from 2.17 to 10.2. The patch size of the proposed MPSA was reduced by about 44.9%, compared to the conventional MPSA, which makes it promising for a compact permittivity sensor. Good agreement between the extracted and the reference relative permittivity with a maximum error less than 2.68% was achieved, which validates the effectiveness of the proposed MPSA.

The possible applications of the proposed high-sensitivity MPSA are permittivity characterization of planar solid materials with low relative permittivity and low loss; proximity detection in automation applications; non-invasive determination of moisture content in soil, food, or liquids; and wireless sensing of biological samples or poisonous chemicals. It can also be applicable as a simultaneous communication-and-sensing unit for chipless radio frequency identification (RFID) sensor tags measuring temperature, humidity, and other characteristics, and communicating with a reader at the same time. 

In future work, we will try to characterize high relative permittivity and high loss materials using the proposed MPSA and compare its sensitivity with other MPSAs.

## Figures and Tables

**Figure 1 sensors-19-04660-f001:**
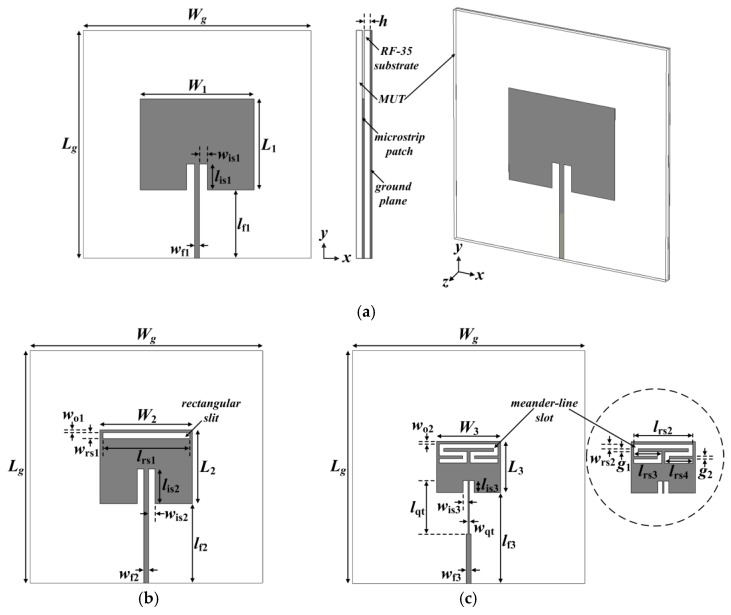
Geometries of the three microstrip patch sensor antennas (MPSAs): (**a**) conventional inset-fed rectangular MPSA; (**b**) rectangular slit (RS)-loaded MPSA; and (**c**) proposed meander-line slot (MLS)-loaded MPSA.

**Figure 2 sensors-19-04660-f002:**
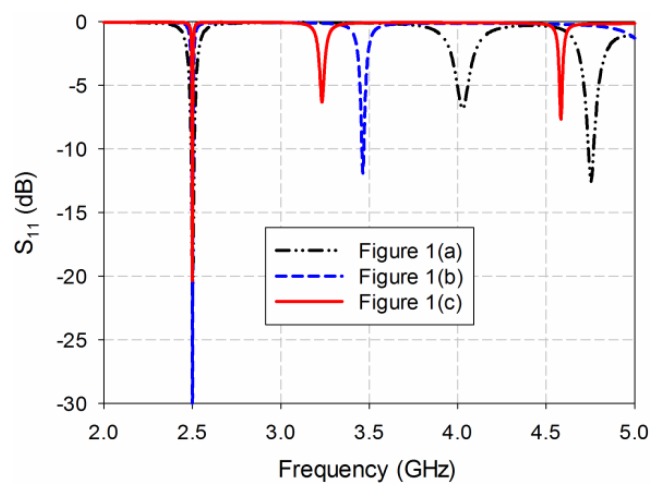
S_11_ characteristics of the three MPSAs in Figure 1.

**Figure 3 sensors-19-04660-f003:**
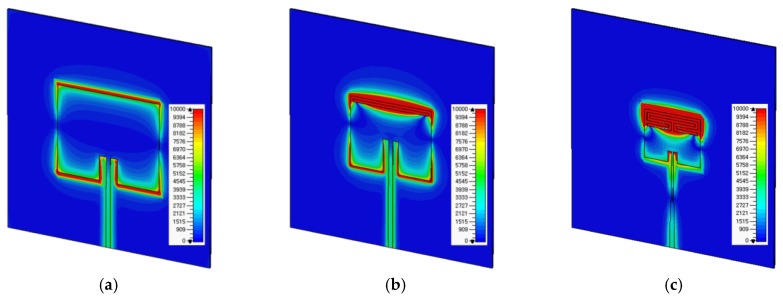
Electric field distributions at 2.5 GHz for (**a**) conventional MPSA; (**b**) RS-loaded MPSA; and (**c**) proposed MLS-loaded MPSA.

**Figure 4 sensors-19-04660-f004:**
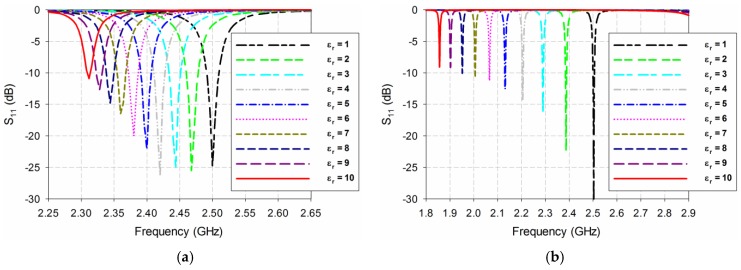
S_11_ characteristics for varying relative permittivity of the material under test (MUT): (**a**) conventional MPSA; (**b**) RS-loaded MPSA; and (**c**) proposed MLS-loaded MPSA.

**Figure 5 sensors-19-04660-f005:**
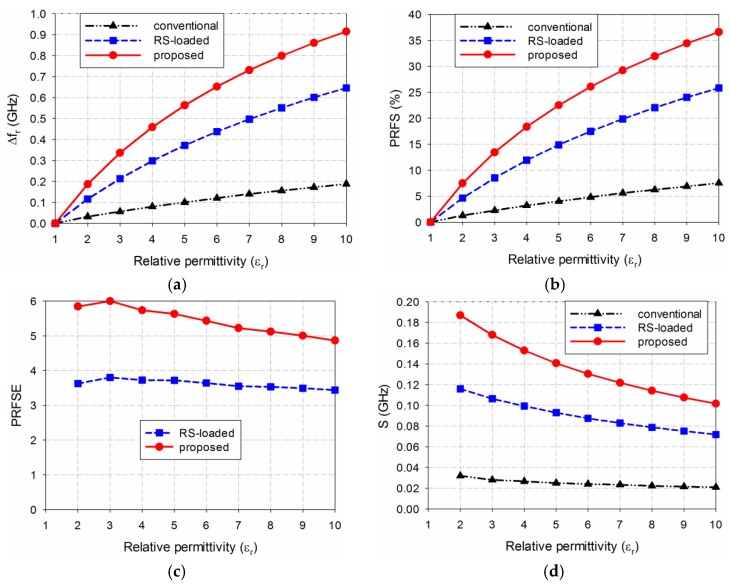
Sensitivity comparison of the three MPSAs: (**a**) Δ*f*_r_; (**b**) PRFS; (**c**) PRFSE; (**d**) S; and (**e**) SE.

**Figure 6 sensors-19-04660-f006:**
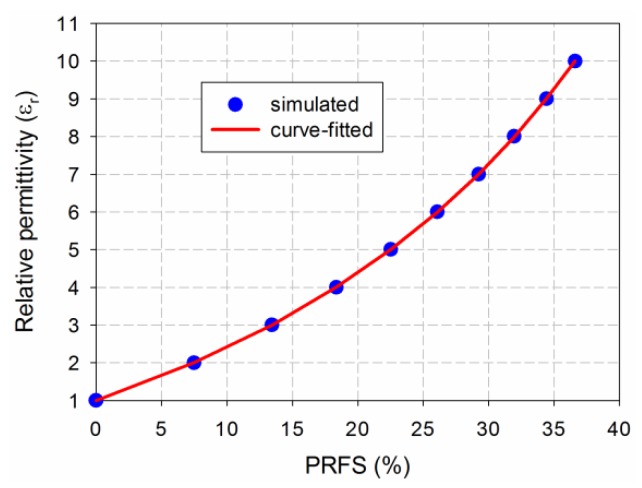
Simulated and curve-fitted relative permittivity of the MUT as a function of PRFS for the proposed MLS-loaded MPSA.

**Figure 7 sensors-19-04660-f007:**
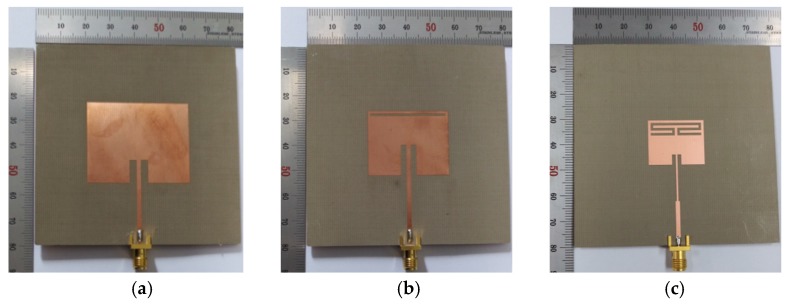
Photographs of the fabricated MPSAs: (**a**) conventional MPSA; (**b**) RS-loaded MPSA; and (**c**) proposed MLS-loaded MPSA.

**Figure 8 sensors-19-04660-f008:**
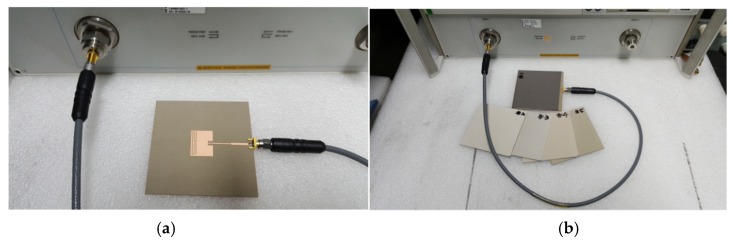
Experiment setups: (**a**) unloaded and (**b**) loaded.

**Figure 9 sensors-19-04660-f009:**
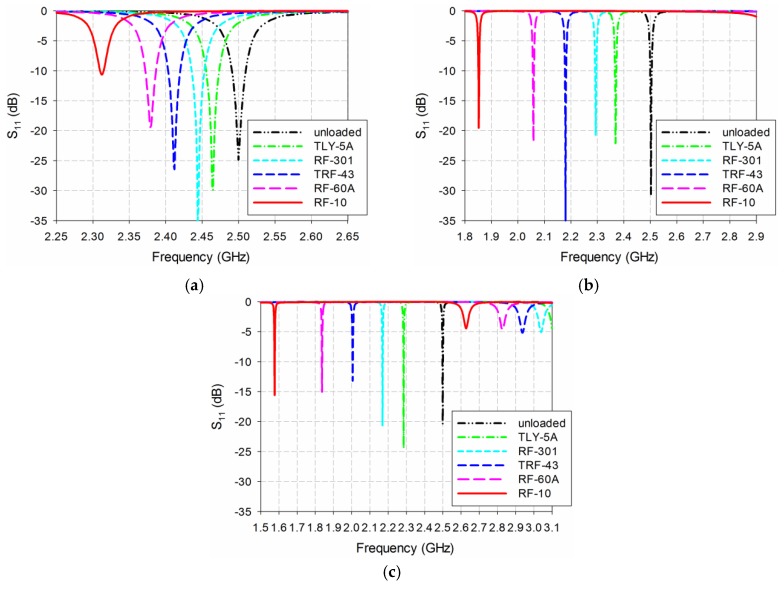
Simulated S_11_ characteristics of the three MPSAs for MUTs in Table 2: (**a**) conventional MPSA; (**b**) RS-loaded MPSA; and (**c**) proposed MLS-loaded MPSA.

**Figure 10 sensors-19-04660-f010:**
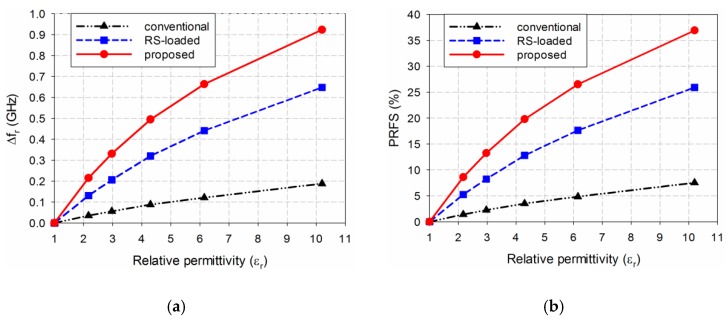
Simulated sensitivity comparison of the three MPASs for the MUTs in Table 2: (**a**) Δ*f*_r_; (**b**) PRFS; (**c**) PRFSE; (**d**) S; and (**e**) SE.

**Figure 11 sensors-19-04660-f011:**
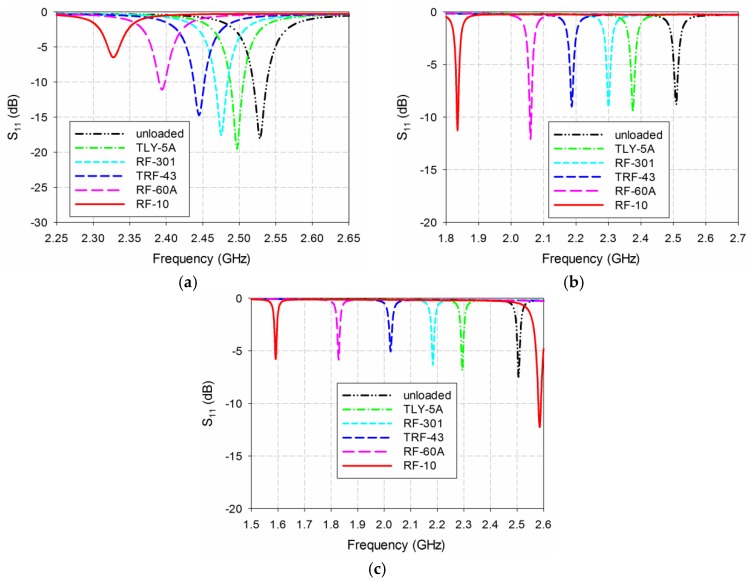
Measured S_11_ characteristics of the three MPSAs for the MUTs in Table 2: (**a**) conventional MPSA; (**b**) RS-loaded MPSA; and (**c**) proposed MLS-loaded MPSA.

**Figure 12 sensors-19-04660-f012:**
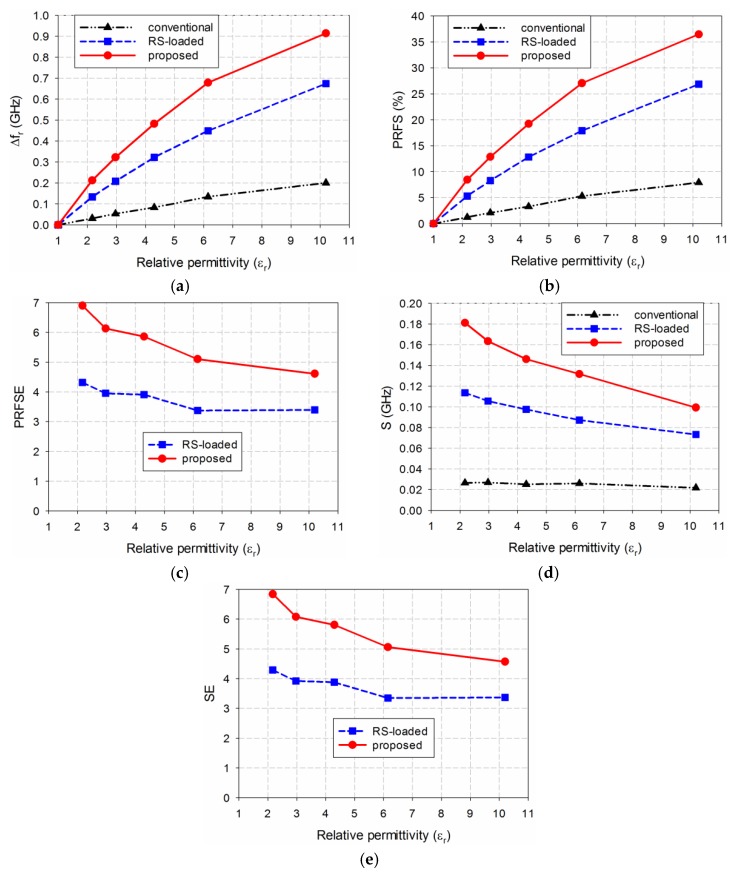
Measured sensitivity comparison of the three MPSAs for the MUTs in Table 2: (**a**) Δ*f*_r_; (**b**) PRFS; (**c**) PRFSE; (**d**) S; and (**e**) SE.

**Table 1 sensors-19-04660-t001:** First resonant frequencies of S_11_ responses of the conventional, RS-loaded, and proposed MLS-loaded MPSAs in GHz.

Antenna Type	*ε*_r_ = 1	*ε*_r_ = 2	*ε*_r_ = 3	*ε*_r_ = 4	*ε*_r_ = 5	*ε*_r_ = 6	*ε*_r_ = 7	*ε*_r_ = 8	*ε*_r_ = 9	*ε*_r_ = 10
Conventional	2.5	2.468	2.444	2.420	2.400	2.380	2.360	2.344	2.328	2.312
RS-loaded	2.5	2.384	2.287	2.202	2.128	2.063	2.003	1.949	1.899	1.854
Proposed	2.5	2.313	2.164	2.041	1.937	1.848	1.769	1.701	1.6395	1.585

**Table 2 sensors-19-04660-t002:** Relative permittivity, loss tangent, and thickness of the five MUTs.

No.	MUT	ε_r_	tan δ	Thickness
1	TLY-5A	2.17 ± 0.02	0.0009 @ 10GHz	1.575 mm
2	RF-301	2.97 ± 0.07	0.0012 @ 1.9GHz	1.524 mm
3	TRF-43	4.3 ± 0.15	0.0035 @ 10GHz	1.626 mm
4	RF-60A	6.15 ± 0.15	0.0028 @ 10GHz	1.524 mm
5	RF-10	10.2 ± 0.3	0.0025 @ 10GHz	1.524 mm

**Table 3 sensors-19-04660-t003:** Simulated first resonant frequencies for S_11_ responses in the three MPSAs in GHz.

Antenna Type	Unloaded (*ε*_r_ = 1)	TLY-5A (*ε*_r_ = 2.17)	RF-301 (*ε*_r_ = 2.97)	TRF-43 (*ε*_r_ = 4.3)	RF-60A (*ε*_r_ = 6.15)	RF-10 (*ε*_r_ = 10.2)
Conventional	2.5	2.465	2.444	2.412	2.379	2.312
RS-loaded	2.5	2.369	2.294	2.18	2.059	1.852
Proposed	2.5	2.285	2.169	2.005	1.837	1.577

**Table 4 sensors-19-04660-t004:** Measured first resonant frequencies of S_11_ responses for the three MPSAs in GHz.

Antenna Type	Unloaded (*ε*_r_ = 1)	TLY-5A (*ε*_r_ = 2.17)	RF-301 (*ε*_r_ = 2.97)	TRF-43 (*ε*_r_ = 4.3)	RF-60A (*ε*_r_ = 6.15)	RF-10 (*ε*_r_ = 10.2)
Conventional	2.528	2.497	2.475	2.445	2.394	2.328
RS-loaded	2.509	2.376	2.301	2.187	2.060	1.835
Proposed	2.506	2.294	2.184	2.024	1.828	1.592

**Table 5 sensors-19-04660-t005:** Comparison of extracted relative permittivity for the five MUTs.

No.	MUT	Reference ε_r_	Extracted ε_r_	Error (%)
1	TLY-5A	2.17 ± 0.02	2.1503	0.91
2	RF-301	2.97 ± 0.07	2.8903	2.68
3	TRF-43	4.3 ± 0.15	4.1983	2.36
4	RF-60A	6.15 ± 0.15	6.2929	−2.32
5	RF-10	10.2 ± 0.3	9.9382	2.57

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
