# Peer review of "Meander-Line Slot-Loaded High-Sensitivity Microstrip Patch Sensor Antenna for Relative Permittivity Measurement"

_sensors, 2019, doi:10.3390/s19214660_

Round 1

Reviewer 1 Report

The proposed work presents a microstrip patch antenna with meander-line-loaded used as a sensor for relative permittivity measurement. The proposed sensor claimed to improve the sensitivity for the relative permittivity measurement ranging from 2.17 to 10.2. However, the theory and design concept are not well described. Furthermore, poor design of the antenna can be observed from the measured S11 for the unloaded condition.  

Abstract

            -As a sensor, the size comparison should be related to the size or requirement of material or object being tested. Typically, the sample is material is larger than sensor. “The proposed MPSA has the advantage of a compact sensing area because 22 the size of the patch was reduced about 44.9% compared to the conventional MPSA” So, this need to be clarified and revised.

Introduction

            -The introduction of relative permittivity measurements should be first introduced in the first paragraph rather than microstrip patch antenna application. Then, the advantage of the microstrip for relative permittivity compares to other measurement techniques should be given.

            -Why the proposed work chooses the meander-line slot (MLS) loaded? The literature of MLS should be given.

            -Please replace all “permittivity” in the paper with “relative permittivity”.

MLS-loaded MPSA design

            -It is possible to explain or describe the MLS as a lumped model or equivalent circuit?

            -Relationship/Equation between the physical dimension of MLS and the resonant frequency or the quality factor should be given.

            -Please see line 134 to 136 “Note that the location of the RS was chosen because the electric field concentration along the radiating edge of the patch and the permittivity sensitivity related to the capacitive perturbation caused by the electric field interaction with the dielectric MUT are highest at this position.” The capacitive perturbation needs to be clarified.

Sensitivity comparison

            -Fig.4, the proposed work mentioned about the “lossless case”. This should be noted that the proposed work should be clarified about the interested material properties/applications in the introduction.

            - Please see lines 208 to 210 “It is well-known that when the MUT is loaded above the patch, it affects the total capacitance and effective relative permittivity related to the microstrip patch resonator, and the S11 resonant frequency is a nonlinear function of the resonator related to the effective relative permittivity.” The total capacitance needs to be clarified by using the equivalent circuit or admittance model.

Experiment results and discussion

            -Please check Fig.8, It look similar to the previous work had been done by the authors.

-In Fig. 11 (c), the Mag of S11 for unloaded condition is very poor, magnitude of S11 < -10dB. This is an issue.

Conclusion

            -The proposed work didn’t demonstrate for high relative permittivity measurement and high dielectric loss materials. I think mentioned about the proposed work suitable for microfluidic liquid materials cannot be accepted.

Author Response

Thanks you a lot for your comments.

Attached pls. find a point-by-point response to the reviewer’s comments.

Best regards,

Reviewer 2 Report

Paper titled "Meander-Line Slot-Loaded High-Sensitivity Microstrip Patch Sensor Antenna for Permittivity Measurement" is a good paper for readers and new researchers. However, I have some comments.

Most of the sentences are started with a verb. make no sense in the text. like line no 30, line no 35, and so on in the paper. Figure 1. The picture is not clear in some sense. the text style is not consistent in the paper. what is the meaning of making equation "BOLD" The patch antenna gain and relative power. Do not u think it's needed to measure as well.  Does it was not good to perform some HFSS results?  The results are very well presented and giving very relative measurements. However, what is the range of antenna sensitivity in Ghz.  from line 358 to line 385 . this discussion is un necessary. or better if its available in the form of tables.

Author Response

(The authors gave the same response as above.)

Round 2

Reviewer 1 Report

Please see the follow up comments as below

Point 1: oK

Point 2: oK

Point 3: ok

Point 4: ok

Point 5: ok

Point 6: This is not clarified.

Point 7: ok

Point 8: ok

Point 9: ok

Point 10: ok

Point 11: The Mag of S11 for unloaded condition is very poor, magnitude of S11 < -10dB. This is an issue.

Point 12: ok

Author Response

Thanks you a lot for your comments.

Attached pls. find the responses to points 6 and 11 of your comments.

Best regards,
